# 12-Lead ECG Generation via a PDE-Based GAN

## Abstract

Synthesizing realistic 12-lead electrocardiogram (ECG) data is a complex task due to the intricate spatial and temporal dynamics of cardiac electrophysiology. Traditional generative models often struggle to capture the nuanced interdependencies among ECG leads, which are essential for accurate medical analysis. In this paper, we introduce a novel method that integrates partial differential equations (PDEs) into a generative adversarial network (GAN) framework to model the spatiotemporal behavior of the heart's electrical activity. By embedding PDE-based representations directly into the generative process, our approach effectively captures both the temporal evolution and spatial relationships between ECG leads. This results in the production of high-fidelity synthetic 12-lead ECG data that closely mirrors real physiological signals. We conduct extensive experiments to evaluate the efficacy of our PDECGAN model, demonstrating that classifiers trained on our synthetic data outperform those trained on data generated by conventional methods in detecting cardiac abnormalities, with statistically significant improvements. Our work highlights the potential of combining PDE-driven cardiac models with advanced generative techniques to enhance the quality and utility of synthetic biomedical datasets.

## 1 Introduction

Electrocardiograms (ECGs) are essential diagnostic tools for monitoring and detecting heart conditions by recording the electrical activity of the heart through multiple leads. The 12-lead ECG, in particular, provides comprehensive spatial information about cardiac function, making it invaluable for identifying a wide range of cardiac abnormalities.

Developing robust machine learning models for ECG analysis requires large and diverse datasets. However, acquiring such datasets poses significant challenges due to privacy concerns, data security issues, and the scarcity of data representing rare cardiac conditions (Voigt & Bussche, 2017). These limitations hinder research and development efforts in cardiac healthcare applications.To overcome these challenges, synthetic data generation has emerged as a promising solution (Giuffrè & Shung, 2023; de Melo et al., 2022). While existing generative models, such as Generative Adversarial Networks (GANs), have made progress, they often focus on single-lead ECGs, neglecting the complex interdependencies and spa- tiotemporal dynamics of 12-lead ECG signals. Accurately modeling these relationships is crucial for generating synthetic data that mirrors real-world cardiac function.

Recent advances in Physics-Informed Neural Networks (PINNs) have shown how integrating physical laws into neural networks can enhance the accuracy of predictions in various scientific domains (Raissi et al., 2019; Karniadakis et al., 2021). PINNs leverage known physical principles, such as partial differential equations (PDEs), to guide the learning process, ensuring that the generated outputs respect the underlying laws of nature. This approach is particularly relevant in our work, where accurately capturing the spatiotemporal dynamics of cardiac electrophysiology is critical. Unlike ordinary differential equations (ODEs), which capture temporal dynamics alone, PDEs model both temporal evolution and spatial relationships.

Similar to PINNs, our method ensures that the synthetic ECGs not only look realistic but also adhere to the physiological principles governing heart activity, represent the intricate interactions within the heart's electrical field and the interdependencies among the 12 ECG leads.

Inspired by this, we propose PDECGAN, a novel generative framework that integrates partial differential equations (PDEs) representing cardiac electrophysiology into the GAN architecture (Good-

fellow et al., 2014). By embedding PDE-based constraints within the generative model, PDECGAN ensures that the synthetic ECG data not only appear realistic but also align with the underlying physiological principles governing cardiac electrical activity.

This integration of PDE-based constraints into the GAN's training process enables the model to learn from both empirical data and the fundamental principles of cardiac electrophysiology, enhancing the realism and accuracy of the synthetic ECG signals. Consequently, the generated data better captures the observed patterns and physiological mechanisms inherent in real ECGs.

Classifiers trained on PDECGAN-generated data demonstrate improved performance in detecting cardiac abnormalities, highlighting the practical advantages of incorporating physiological models into the data generation process. Our empirical analysis confirms the effectiveness of this approach in boosting classifier accuracy for heart anomaly detection.

Our key contributions in this study are as follows:

1. **The PDECGAN Framework:** We introduce PDECGAN, a novel generative framework that integrates partial differential equations into a Generative Adversarial Network to accurately model the spatiotemporal dynamics of 12-lead ECG signals.

2. **Enhanced Synthetic Data Quality:** By leveraging PDEs, PDECGAN generates high-fidelity synthetic 12-Lead ECG heartbearts data that captures the complex relationships between leads.

3. **Improved Classification Performance:** Classifiers trained on synthetic data generated by PDECGAN demonstrate superior performance in detecting cardiac abnormalities compared to those trained on data from traditional generative models.

This work showcases how integrating physiological modeling with machine learning can capture complex cardiac dynamics, akin to modeling natural phenomena, enhancing the accuracy and effectiveness of diagnostic models in medical applications.

## 2 RELATED WORK

The success of deep learning (DL) models, particularly in medical applications, hinges on the availability of large, annotated datasets. However, data scarcity, privacy concerns, and ethical constraints, especially in the medical domain, hinder the collection of such datasets (de Melo et al., 2022). To address these limitations, synthetic data generation has emerged as a crucial tool, providing high-quality, diverse datasets for training deep neural networks (DNNs) (Cichy & Kaiser, 2019).

In the realm of cardiovascular diagnostics, synthetic electrocardiogram (ECG) data has gained attention as a means to improve diagnostic models. ECGs are critical for diagnosing various heart conditions, and generating realistic synthetic ECGs facilitates the development of robust machine learning models.

Generative Adversarial Networks (GANs) (Goodfellow et al., 2014) have significantly advanced synthetic data generation,including ECGs (Donahue et al., 2018). Early applications of GANs in ECG generation, such as those by Zhu et al. (2019), Golany et al. (2020) and Golany et al. (2021), demonstrated the potential of adversarial models, though they were primarily focused on single-lead ECG generation, limiting their ability to capture the complexities of 12-lead signals.

More recent efforts have sought to address this limitation. Several methods have been proposed for generating 12-lead ECG data. For instance, Liu et al. (2020) utilized vector quantized variational autoencoders (VQ-VAE) to improve ECG classifiers. Huang et al. (2023) developed a GAN-based model for unsupervised noise ECG generation, and Alcaraz & Strodthoff (2023) presented SSSD-ECG, a diffusion-based technique for 12-lead ECG synthesis. The most recent advancement, MultiODE-GAN (Yehuda & Radinsky, 2024), extends ODE-based methods by simulating heart activity with coupled ODEs, building on the works of (Golany et al., 2020) and (McSharry et al., 2003). In our study, we compare PDECGAN with the current state-of-the-art models, specifically Alcaraz & Strodthoff (2023) and (Yehuda & Radinsky, 2024).

Our work builds on these advancements by integrating partial differential equations (PDEs) with GANs to model the spatiotemporal dynamics of 12-lead ECGs. PDEs provide a more comprehen-

sive simulation of heart activity, capturing both the temporal progression and spatial distribution of electrical signals across the heart. This approach is inspired by Physics-Informed Neural Networks (PINNs) (Raissi et al., 2019; Karniadakis et al., 2021), which have shown the effectiveness of embedding physical laws, such as PDEs, into neural networks. By incorporating these principles, PDECGAN achieves state-of-the-art performance in generating high-fidelity 12-lead ECG data, surpassing existing methods.

Furthermore, the synthetic data generated by PDECGAN demonstrates practical utility in improving the performance of 12-lead ECG classifiers, showing enhanced results when trained on a combination of real and synthetic data (Ribeiro et al., 2020).

## 3 METHOD

In this section, we present our approach for integrating partial differential equations (PDEs) into a Generative Adversarial Network (GAN) framework to generate realistic 12-lead electrocardiogram (ECG) heartbeats. Our method leverages the strengths of PDEs in modeling the spatiotemporal dynamics of cardiac electrical activity and the generative capabilities of GANs.

### 3.1 MOTIVATION AND PDE FORMULATION

The heart's electrical activity is inherently spatiotemporal, with electrical impulses propagating across cardiac tissue. PDEs naturally capture these dynamics, modeling both spatial and temporal variations in a unified framework. By formulating the ECG generation process as a PDE, we incorporate physiological parameters that reflect realistic cardiac dynamics.

$$\frac{\partial \mathbf{u}}{\partial t} = F(\mathbf{u}, \mathbf{x}, t, \nabla \mathbf{u}, \nabla^2 \mathbf{u}, \ldots, \boldsymbol{\theta}), \tag{1}$$

Here, $\mathbf{u}(t, \mathbf{x})$ represents the multi-dimensional ECG signal at time $t$ and spatial coordinates $\mathbf{x}$, while $F$ defines the dynamics of the ECG signals, including physiological parameters $\boldsymbol{\theta}$ such as heart rate variability and ECG morphology.

Our proposed framework simulates the dynamics of 12-lead ECG heartbeats using a system of PDEs that capture both temporal and spatial variations inherent in cardiac electrical activity. The PDE for each lead $i$ becomes:

$$\frac{\partial u_i(t)}{\partial t} = f(u_i(t); \theta) + D \sum_{j \in \mathcal{N}(i)} w_{ij}[u_j(t) - u_i(t)], \tag{2}$$

where:

- $u_i(t)$ is the ECG signal amplitude at lead $i$ and time $t$.
- $f(u_i(t); \theta)$: Nonlinear function representing physiological processes, modeled by a neural network with parameters $\theta$.
- $D$ Diffusion coefficient modeling spatial interactions between leads.
- $\mathcal{N}(i)$: Set of neighboring leads to lead $i$.
- $w_{ij}$: Weighting factor representing the interaction strength between leads $i$ and $j$.

### 3.2 NUMERICAL SOLUTION

#### 3.2.1 TEMPORAL DERIVATIVE APPROXIMATION

We approximate the temporal derivative of the ECG signal using finite differences. The temporal derivative for lead $i$ at time $t$ is approximated using a forward difference:

$$\frac{\partial u_i(x, t)}{\partial t} \approx \frac{u_i(x, t + \Delta t) - u_i(x, t)}{\Delta t} \tag{3}$$

where:

- $u_i(x, t)$: Signal amplitude at spatial position $x$ and time $t$.
- $\Delta t$ is the sampling interval.

### 3.2.2 SPATIAL INTERACTION APPROXIMATION

The spatial interaction term in equation 2 is approximated using:

$$\sum_{j \in \mathcal{N}(i)} w_{ij}[u_j(t) - u_i(t)], \tag{4}$$

where $w_{ij}$ is determined based on the physical proximity and physiological relationships between leads $i$ and $j$.

### 3.2.3 INITIAL CONDITIONS AND PDESTEP FUNCTION

The generator receives an initial condition $\mathbf{U}(0)$ for all leads, which is generated from a latent vector $z$ sampled from a prior distribution (e.g., Gaussian).

$$\mathbf{U}(0) = G(z; \theta_G), \tag{5}$$

where $\theta_G$ are the parameters of the generator network.

To simulate the temporal evolution of the ECG signals, we iteratively apply a function called `PDEStep`, which advances the ECG signals from time $t$ to $t + \Delta t$:

$$\mathbf{U}(t + \Delta t) = \text{PDEStep}(\mathbf{U}(t), \Delta t, G(\mathbf{U}(t), \mathbf{x}_t, t, z; \theta_G)), \tag{6}$$

In this context, `PDEStep` computes the next state of the ECG signals using the discretized PDEs, and $G$ provides the necessary neural network computations for $f(u_i(t); \theta)$.

### 3.2.4 NUMERICAL INTEGRATION AND SOLVE FUNCTION

The entire sequence of ECG signals over the time interval $[0, T]$ is obtained by iteratively applying the `PDEStep` function. This process is represented by a `Solve` function:

$$\mathbf{U} = [\mathbf{U}(0), \mathbf{U}(\Delta t), \dots, \mathbf{U}(T)] = \text{Solve}(\mathbf{U}(0); \theta_G), \tag{7}$$

where $\theta_G$ represents the generator's parameters.

The `Solve` function effectively integrates the PDE over time, using the initial condition and generator's dynamics, to produce the complete ECG signal sequence.

### 3.2.5 UPDATE RULE

Combining Equations equation 2 and the finite difference approximation of the temporal derivative, the update rule within the `PDEStep` function is:

$$u_i(t + \Delta t) = u_i(t) + \Delta t \left[ f(u_i(t); \theta) + D \sum_{j \in \mathcal{N}(i)} w_{ij}[u_j(t) - u_i(t)] \right], \tag{8}$$

This equation is used within the `PDEStep` function to advance the ECG signals at each time step.

## 3.3 GAN FRAMEWORK

### 3.3.1 GENERATOR AND DISCRIMINATOR DESIGN

**Generator** $G$    The generator aims to produce ECG signals that satisfy both the data distribution and the PDE constraints. It receives the initial condition $\mathbf{U}(0)$ from Equation equation 5 and computes the entire sequence $\mathbf{U}$ by iteratively applying the `PDEStep` function within the `Solve` process as described in Equation equation 7.

**Discriminator** $D$    The discriminator distinguishes between real ECG signals and those generated by $G$. It assesses the realism of the entire 12-lead ECG signal sequence.

### 3.3.2 LOSS FUNCTIONS

**Adversarial Loss**    The adversarial loss for the generator is defined as:

$$L_{\text{adv}}^{(G)} = -\mathbb{E}_{z \sim p_z} \left[ \log D(\hat{\mathbf{U}}) \right],$$ (9)

where $\hat{\mathbf{U}} = \texttt{Solve}(G(z; \theta))$ is the generated ECG signal sequence.

The adversarial loss for the discriminator is:

$$L_{\text{adv}}^{(D)} = -\mathbb{E}_{\mathbf{U} \sim p_{\text{data}}} [\log D(\mathbf{U})] - \mathbb{E}_{z \sim p_z} \left[ \log(1 - D(\hat{\mathbf{U}})) \right],$$ (10)

where $\mathbf{U}$ represents real ECG signal sequences from the dataset.

**PDE Loss**    To ensure that the generated signals satisfy the PDE constraints, we define a PDE loss based on the residual of the discretized PDE within each `PDEStep`:

$$\text{Residual}_i(t) = \frac{u_i(t + \Delta t) - u_i(t)}{\Delta t} - \left[ f(u_i(t); \theta_f) + D \sum_{j \in \mathcal{N}(i)} w_{ij}[u_j(t) - u_i(t)] \right].$$ (11)

The PDE loss is then defined as:

$$L_{\text{PDE}} = \frac{1}{N} \sum_{i=1}^{N} \sum_{t} \left( \text{Residual}_i(t) \right)^2,$$ (12)

where $N$ is the number of leads.

**Total Generator Loss**    The total loss for the generator combines the adversarial and PDE losses:

$$L_G = \lambda_{\text{adv}} L_{\text{adv}}^{(G)} + \lambda_{\text{PDE}} L_{\text{PDE}},$$ (13)

where $\lambda_{\text{adv}}$ and $\lambda_{\text{PDE}}$ are hyperparameters that balance the importance of each loss term.

## 4 ECG CLASSIFIER

To evaluate the realism and utility of the synthetic 12-lead ECG data generated by PDECGAN, we employ a state-of-the-art Residual Neural Network (ResNet) architecture, which is well-established for its effectiveness in 12-lead ECG classification tasks (Attia et al., 2019; Ribeiro et al., 2020; Nejedly et al., 2021). This model serves as a benchmark for assessing the impact of synthetic data by comparing its performance when trained on real data versus a combination of real and synthetic data.

The ResNet implementation begins with a convolutional layer containing 16 filters of size 7x7, followed by a max-pooling layer to reduce dimensionality while enhancing feature extraction. The network's backbone consists of five residual blocks, each made up of three convolutional layers with batch normalization and ReLU activation. Each block incorporates a skip connection, ensuring that input data bypasses the convolutional layers and connects directly to the block's output. This structure mitigates the degradation of training signals, allowing for deeper network training.

The network complexity increases across blocks, with the number of filters growing from 16 in the first block to 64 in the last. A stride of 2 is applied to the first convolutional layer of each block (except the first) to downsize the temporal dimension and manage computational load. The final global average pooling layer reduces the feature space, feeding into a dense layer that outputs the classification probabilities for abnormal ECGs, using a sigmoid activation function for binary classification.

To ensure optimal training, the network's weights are initialized using the initialization method (He et al., 2016), while biases are set to zero to maintain consistency and stability during training.

## 5 EXPERIMENTAL EVALUATION

### ECG DATASET

The empirical evaluation of our proposed method relies on the use of the Georgia 12-Lead ECG Challenge (G12EC) dataset, sourced from Emory University, Atlanta, Georgia, as part of the Physionet 2020 Challenge (Alday et al., 2020). This dataset is pivotal to our study due to its comprehensive coverage and diverse representation of cardiac conditions. It consists of 10,344 12-lead ECG recordings obtained from 7,871 patients, reflecting a broad demographic spectrum primarily from the southeastern United States.

Each recording in the G12EC dataset captures 10 seconds of ECG data, sampled at a high resolution of 500 Hz, translating to 5,000 time points per recording. This high-frequency data collection ensures detailed ECG waveforms, facilitating accurate analysis and synthesis in our research.

To ensure robust model evaluation, we employed 5-fold cross-validation, dividing the dataset into training and validation subsets. This approach allows for more reliable performance estimates by rotating the training and validation data across different splits. In addition, 20% of the dataset was set aside as an independent test set, reserved exclusively for evaluating the final model's performance. This setup ensures that the model assessment is rigorous and generalizes well to unseen data.

### EXPERIMENTAL METHODOLOGY

To evaluate the quality of the synthetic data generated by PDECGAN, we follow the evaluation methodology outlined in (Alcaraz & Strodthoff, 2023; Yehuda & Radinsky, 2024), focusing on 12-lead ECG classification using the G12C dataset.

We train a ResNet classifier (Section 4 ) on both real and synthetic data, then assess its performance on a real test set. Comparable performance between models trained solely on real data and those trained on a combination of real and synthetic data would indicate the synthetic data's high quality. We evaluate three scenarios:

**No synthetic data:** Training is conducted exclusively on real data.

**Synthetic data from PDECGAN:** Training combines real data with synthetic ECG heartbeats generated by PDECGAN.

**Synthetic data from other models:**Training uses real data and synthetic ECGs generated by alternative models (Section 6).

If the model trained on a combination of real and synthetic data outperforms the model trained solely on real data, it indicates that the synthetic data is of high quality and contributes positively to model training. Conversely, a noticeable drop in test accuracy would suggest that the synthetic data deviates from the real data distribution, reflecting lower quality in the generated samples.

IMPLEMENTATION DETAILS

To prepare the raw ECG data for training and testing, we applied signal processing techniques using NeuroKit2, a Python library for neurophysiological signal processing (Makowski et al., 2021). The ECG signal was segmented into individual heartbeat cycles by detecting R-peaks, which indicate ventricular contractions. These R-peaks were detected using Lead II, known for its clear peak visibility, and the RR intervals (time between successive R-peaks) were used for segmentation.

This segmentation is initially performed on Lead II, commonly used due to its clear visibility of R-peaks, and serves as a reference for synchronizing the segmentation across the remaining leads. By aligning the segmentation of all 12 leads with that of Lead II, we ensure temporal coherence across the dataset, maintaining the integrity of the cardiac cycle's representation in each lead.

Additionally, to enhance the relevance and reliability of our dataset for training generative models, we specifically select ECG examples that exhibit abnormalities consistently across all cycles. This approach avoids the potential bias and variance in training that might arise from using examples where anomalies are only present in isolated cycles within the signal. Each segmented ECG cycle, represented as $\hat{x} \in \mathbb{R}^{12 \times L}$, retains the original annotations from the full signal $x \in \mathbb{R}^{12 \times 5000}$, ensuring that each cycle is accurately labeled according to the comprehensive diagnostic features of the overall ECG recording.

## 6 EXPERIMENTAL RESULTS

MAIN RESULT

This study evaluates the effectiveness of PDECGAN, a generative model that uses partial differential equations (PDEs) to produce high-quality synthetic 12-lead ECG data. We assess its impact by comparing the sensitivity and specificity of heart abnormality detection between classifiers trained solely on real data and those trained on a combination of real and synthetic data.

As detailed in Table 1, incorporating synthetic data significantly improves specificity (while maintaining constant sensitivity) across various heart abnormalities. Classifiers trained on both real and synthetic data consistently outperformed those trained on real data alone, demonstrating the synthetic data's role in enhancing model robustness and diagnostic accuracy. The addition of synthetic data reduces overfitting and improves generalization to unseen real-world data.

The improvements were validated using a 5-fold cross-validated paired t-test, with p-values below 0.05, confirming their statistical significance. Significant improvements are highlighted in bold in the table.

Table 1: Baseline Classifier Performance: Real Data vs. Real + PDECGAN Synthetic Data

| Abnormality | Baseline CLS(Ribeiro et al.) | | PDECGAN | |
|---|---|---|---|---|
| | Sensitivity | Specificity | Sensitivity | Specificity |
| IAVB | 0.94 | 0.82 | 0.94 | **0.87** |
| RBBB | 0.94 | 0.89 | 0.94 | **0.93** |
| LBBB | 0.97 | 0.96 | 0.97 | 0.96 |
| NSIVCB | 0.78 | 0.72 | 0.78 | **0.79** |
| LAnFB | 0.89 | 0.76 | 0.89 | **0.82** |
| LAD | 0.89 | 0.88 | 0.89 | **0.91** |
| QAb | 0.81 | 0.70 | 0.81 | **0.74** |
| AFL | 0.93 | 0.83 | 0.93 | **0.88** |

COMPARATIVE ANALYSIS OF GENERATIVE MODELS

In this ablation study, we compare the performance of various generative models in producing synthetic 12-lead ECG data, particularly focusing on how these models influence the specificity of classifiers trained with the generated data. The objective is to highlight the strengths of different

generative approaches and demonstrate the superior capability of our proposed method, PDECGAN, in enhancing classifier performance. We included several generative models in our analysis, with the latest being state-of-the-art approaches:

- DCGAN (Radford et al., 2016): A GAN variant known for stable training dynamics and success in image generation, adapted here for waveform data.
- WaveGAN (Donahue et al., 2018): Designed specifically for generating audio waveforms, WaveGAN effectively captures temporal patterns and is applicable to ECG data.
- SSSD-ECG (Alcaraz & Strodthoff, 2023): A recent diffusion-based model designed for generating synthetic multi-lead ECG data.
- MultiODE-GAN (Yehuda & Radinsky, 2024): An ODE-based generative model that refines ECG signal realism by simulating heart activity with coupled ODEs.
- PDECGAN: Our proposed method that integrates PDEs into the generative process to encapsulate both spatial and temporal heart dynamics comprehensively.

Table 2: Classifier Performance with Real and Synthetic Data from Various Generative Models

| Abnormality | Sensitivity | Specificity | | | | |
|---|---|---|---|---|---|---|
| | | DCGAN | WaveGAN | SSSD-ECG | MultiODEGAN | PDECGAN |
| IAVB | 0.94 | 0.82 | 0.83 | 0.84 | 0.85 | **0.87** |
| RBBB | 0.94 | 0.89 | 0.89 | 0.90 | 0.92 | **0.93** |
| LBBB | 0.97 | 0.95 | 0.96 | 0.96 | 0.96 | 0.96 |
| NSIVCB | 0.78 | 0.70 | 0.73 | 0.75 | 0.76 | **0.79** |
| LAnFB | 0.89 | 0.76 | 0.77 | 0.77 | 0.80 | **0.82** |
| LAD | 0.89 | 0.87 | 0.89 | 0.90 | 0.91 | 0.91 |
| QAb | 0.81 | 0.70 | 0.70 | 0.72 | 0.72 | **0.74** |
| AFL | 0.93 | 0.82 | 0.84 | 0.85 | 0.87 | **0.88** |

IMPACT OF CLASSIFIER ARCHITECTURES

In this experiment, we explore how different classifier architectures utilize synthetic data generated by PDECGAN. Specifically, we assess whether alternative architectures can better leverage this data.

We compare two models:

- **Standard ResNet** (Ribeiro et al., 2020): Our baseline classifier, shown in Table 1 , used in previous experiments.
- **ResNet with Multi-Head Attention** (Nejedly et al., 2021): An enhanced ResNet incorporating attention mechanisms, which performed highly in the PhysioNet Challenge.

Both classifiers were trained on a combination of real and synthetic data generated by PDECGAN and evaluated on a real test set. As shown in Table 3, the ResNet with Multi-Head Attention demonstrated significantly improved specificity when incorporating synthetic data from PDECGAN, compared to being trained only on real data.

These results emphasize the compatibility of our synthetic data with advanced neural network architectures, showing that PDECGAN not only meets the complexity demands of sophisticated models but also enhances their performance. This is particularly important in medical applications, where diagnostic accuracy is crucial, and real-world data may be limited or imbalanced.

IMPACT OF SYNTHETIC DATA NUMBER

We conducted an experiment to examines how varying the quantity of synthetic samples generated by PDECGAN influences classifier performance in detecting heart abnormalities from 12-lead ECG data.

Table 3: Evaluation Results for a Different Classifier Architecture

| Abnormality | CLS(Nejedly et al.) | | PDECGAN | |
|---|---|---|---|---|
| | Sensitivity | Specificity | Sensitivity | Specificity |
| IAVB | 0.93 | 0.85 | 0.93 | **0.90** |
| RBBB | 0.93 | 0.90 | 0.93 | **0.92** |
| LBBB | 0.96 | 0.96 | 0.96 | **0.97** |
| NSIVCB | 0.80 | 0.73 | 0.80 | **0.78** |
| LAnFB | 0.90 | 0.76 | 0.90 | **0.80** |
| LAD | 0.91 | 0.87 | 0.91 | **0.90** |
| QAb | 0.83 | 0.72 | 0.83 | **0.76** |
| AFL | 0.92 | 0.82 | 0.92 | **0.88** |

**Experimental Design:** We generated synthetic datasets at different scales relative to the number of real samples (N) for each class: 0.3N, 0.7N, N, 1.5N, and 2N. Classifiers were trained on these augmented datasets and evaluated on a fixed real test set, focusing on changes in specificity while maintaining constant sensitivity.

As shown in Table 4, increasing the volume of synthetic data generally improved specificity, with peak performance typically observed at N and 1.5N synthetic samples.

Table 4: Classifier Performance Comparison at Various Sample Sizes

| Abnormality | Sensitivity | Specificity | | | | |
|---|---|---|---|---|---|---|
| | | 0.3N | 0.7N | N | 1.5N | 2N |
| IAVB | 0.94 | 0.83 | 0.84 | **0.87** | **0.87** | 0.85 |
| RBBB | 0.94 | 0.90 | 0.92 | **0.93** | 0.92 | 0.91 |
| LBBB | 0.97 | **0.96** | **0.96** | **0.96** | 0.95 | 0.95 |
| NSIVCB | 0.78 | 0.74 | 0.78 | **0.79** | **0.79** | **0.79** |
| LAnFB | 0.89 | 0.78 | 0.81 | **0.82** | **0.82** | 0.80 |
| LAD | 0.89 | 0.90 | 0.90 | **0.91** | **0.91** | 0.90 |
| QAb | 0.81 | 0.72 | **0.74** | **0.74** | **0.74** | 0.73 |
| AFL | 0.93 | 0.85 | 0.86 | **0.88** | **0.88** | 0.86 |

EFFECT OF SPLITTING THE PDE LOSS

We conducted an experiment to evaluate the individual and combined contributions of the temporal and spatial derivatives within our PDE-GAN framework. This helps quantify the impact of each component on the fidelity and realism of the generated 12-lead ECG signals.

To assess the effects of these constraints separately, we split the PDE loss into temporal and spatial components:

$$L_{\text{PDE}} = \lambda_{\text{PDE}} L_{\text{Temporal}} + (1 - \lambda_{\text{PDE}}) L_{\text{Spatial}}, \tag{14}$$

where:

- $L_{\text{Temporal}} = \frac{1}{N} \sum_{i=1}^{N} \sum_t \left[ \frac{u_i(t+\Delta t) - u_i(t)}{\Delta t} - f(u_i(t); \theta) \right]^2$,

- $L_{\text{Spatial}} = \frac{1}{N} \sum_{i=1}^{N} \sum_t \left[ D \sum_{j \in \mathcal{N}(i)} w_{ij} [u_j(t) - u_i(t)] \right]^2$,

- $\lambda_{\text{PDE}}$ is a hyperparameter that balances between the spatia to the temporal loss component.

By adjusting $\lambda_{\text{PDE}}$, we control the influence of each component:

- $\lambda_{\text{PDE}} = 0$ isolates the spatial constraint, ignoring temporal dynamics.

- $\lambda_{\text{PDE}} = 1$ isolates the temporal dynamics, removing the spatial constraint.

As shown in Figure 1, experiments with various $\lambda_{\text{PDE}}$ values (ranging from 0 to 1) reveal that $\lambda_{\text{PDE}} = 0.7$ achieves optimal classifier performance. This suggests that a balanced approach effectively leverages both temporal and spatial dependencies, with temporal dynamics being crucial but not exclusively dominant.

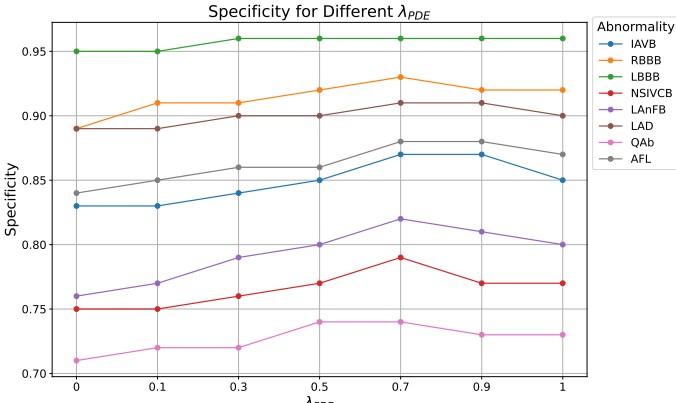

Figure 1: Classifier performance (specificity with constant sensitivity) across different $\lambda_{\text{PDE}}$ value. Optimal performance is achieved at $\lambda_{\text{PDE}} = 0.7$, demonstrating the performance balance between temporal and spatial components.

## 7 CONCLUSION

In this work, we introduced PDECGAN, a novel framework that integrates partial differential equations (PDEs) into a GAN architecture to generate realistic 12-lead ECG signals. By embedding both spatial and temporal constraints, PDECGAN produces physiologically accurate synthetic data.

Our results demonstrate that classifiers trained on a combination of real and PDECGAN-generated data outperform those trained on real data alone, achieving significant improvements in specificity and generalization. This highlights the value of integrating physiological models into generative frameworks for medical data synthesis.

PDECGAN enhances the fidelity of synthetic ECG data, making it highly valuable for clinical research and machine learning applications in healthcare.

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

## A  INCORPORATING ECG LEAD RELATIONSHIPS

The 12-lead ECG system includes standard limb leads (I, II, III) and augmented limb leads (aVR, aVL, aVF). These leads are interrelated based on Einthoven's and Goldberger's laws:

$$\text{Lead I} = \text{Lead II} - \text{Lead III}, \tag{15}$$

$$\text{Lead II} = \text{Lead I} + \text{Lead III}, \tag{16}$$

$$\text{Lead III} = \text{Lead II} - \text{Lead I}, \tag{17}$$

$$aVR = -\frac{1}{2}(\text{Lead I} + \text{Lead II}), \tag{18}$$

$$aVL = \frac{1}{2}(\text{Lead I} - \text{Lead III}), \tag{19}$$

$$aVF = \frac{1}{2}(\text{Lead II} + \text{Lead III}). \tag{20}$$

These relationships imply that not all leads are independent. Specifically, the limb leads can be determined from any two of them.

## B  HYPERPARAMETERS

The hyperparameters $D$, $w_{ij}$, $\lambda_{\text{adv}}$, and $\lambda_{\text{PDE}}$ are tuned to balance the trade-off between signal realism and adherence to physiological dynamics. The diffusion coefficient $D$ and weights $w_{ij}$ are set based on the physical proximity and physiological relationships between the leads.

## C  TRAINING PROCEDURE

We train the generator and discriminator iteratively using the standard GAN training procedure, incorporating the PDE loss to enforce the physical constraints of the ECG dynamics. At each iteration, we:

1. Sample a batch of latent vectors $\{\mathbf{z}_k\}$ and generate synthetic ECG sequences $\{\hat{\mathbf{U}}_k\}$.

2. Compute the adversarial loss $L_{\text{adv}}^{(D)}$ and update the discriminator parameters to minimize this loss.

3. Compute the total generator loss $L_G$ using Equation equation 13.

4. Update the generator parameters $\theta_G$ and $\theta_f$ to minimize $L_G$.

**Role of the Neural Networks**   Two neural networks are involved in our model:

- The generator network $G$, which generates the initial ECG state $\mathbf{u}(0)$ from a latent vector $\mathbf{z}$.

- The network modeling $f(u_i(t); \theta_f)$, which captures the nonlinear physiological processes influencing the ECG signals over time.

**Choice of the Diffusion Coefficient $D$**   The diffusion coefficient $D$ controls the strength of spatial interactions between leads. A larger $D$ implies stronger coupling between leads, which can capture correlated activities across different parts of the heart.

**Determination of Interaction Weights $w_{ij}$**   The weights $w_{ij}$ are determined based on anatomical knowledge and empirical correlations between ECG leads. For instance, leads that are physically close or measure similar cardiac activity will have higher weights.

**Numerical Stability and Time Step $\Delta t$**   The choice of $\Delta t$ is crucial for numerical stability. We select $\Delta t$ to satisfy the Courant–Friedrichs–Lewy (CFL) condition for the discrete system, ensuring stable and accurate simulations.

**Integration with GAN Training**   The PDE constraints are integrated into the GAN training by incorporating the PDE loss $L_{\text{PDE}}$ into the total generator loss. This enforces the physical plausibility of the generated ECG signals while still allowing the generator to learn from the data distribution through the adversarial loss.

