# OpenReview forum: "12-Lead ECG Generation via a PDE-Based GAN"
_ICLR.cc/2025/Conference — ICLR 2025 Conference Withdrawn Submission_

### Official Review · Reviewer_7wUt · 2024-10-23

**Soundness:** 2
**Presentation:** 3
**Contribution:** 2
**Rating:** 3
**Confidence:** 4

**Summary:**

This paper presents a GAN integrated with PDE for better ECG generation. This is achieved by defining a PDE on each lead which describes the temporal dynamics of the ECG generation (described by a neural network) and its spatial relation with neighboring leads. The generation of the PDE is done by solving the PDE with an adversarial loss applied to the generated signal. An additional PDE residual loss is applied. The performance of the model is evaluated by generating synthetic data off the G12EC dataset, and demonstrate whether the addition of synthetic data to real data improves the performance of a classifier.

**Strengths:**

- The idea of integrating a PDE into GAN-baed model is interesting.
- The experimental evaluation considered several relevant baselines, and the presented PDECGAN demonstrated statistically significant improvements in the classifier results.

**Weaknesses:**

1. While the method is motivated by integrating physics into neural networks (such as PINNs), the presented method does not really integrating physics other than the spatial constraints among leads; the major part of the PDE, f(u(t); theta), is an unknown neural network rather than known physics as in a typically PINN.

2. The paper talks about the benefit of a PDE vs. ODE, yet equation (2) is more of a neural ODE with an added constraint among neighboring leads. It is not clear whether the author can claim the benefit brought by a PDE.

3. It is not clear how exactly wij determined in (4) by the vague description on “physical proximity and physiological relationships”. It is especially not clear what kind of “physiological relationship” is being referred to here.

4. It is not clear what is the benefit of the PDE loss in this setting, since this is not a known PDE but a learned neural network in the first portion. It may be helpful to add an ablation on the effect of this term (e.g., completely removing the PDE residual from the loss).

5. It is overall not clear what is the benefit of the temporal terms (modeled by a neural ODE) within equation (2) — it seems to me that the main benefits and innovation of the model comes from the spatial constrain being added to the node. It’d be good to add a vanilla GAN but with this spatial constraint, to demonstrate what is the benefit of the temporal component of equation 2.

6. The paper should discuss and include baselines representing synthetic ECG generation based on physics-based simulation, as there are increasing fast 12-lead ECG simulation pipeline available for such purposes [1]. It is not clear what PDECGAN can achieve that cannot be achieved by these fast physics-based ECG simulation pipelines.

[1] Gillette et al, MedalCare-XL: 16,900 healthy and pathological synthetic 12 lead ECGs from electrophysiological simulations

**Questions:**

1. It'd be helpful if the authors could clarify what specific physical principles, if any, are incorporated into f(u(t); theta).

2. In response to bullet 2 in the weakness section, please provide a more detailed comparison between your PDE-based approach and ODE-based methods, highlighting specific advantages of the proposed method.

3. Please provide a more detailed explanation of how wij in equation 4 are calculated, particularly regarding the physiological relationships.

4. Please add an ablation study to remove the PDE residual from the loss, and report on how it affects the model's performance and the quality of generated ECGs.

5. Please  add a vanilla GAN with the spatial constraint in Equation 2, to demonstrate what is the benefit of the temporal component of Equation 2.

6. Please add baselines representing synthetic ECG generation based on physics-based simulation, and and discuss the specific advantages or limitations of PDECGAN compared to these physics-based approaches.

---

### Official Review · Reviewer_3Fxa · 2024-10-27

**Soundness:** 3
**Presentation:** 2
**Contribution:** 1
**Rating:** 3
**Confidence:** 4

**Summary:**

This paper claims that the proposed model can generate 12-lead electrocardiograms by capturing the nuanced interdependencies among ECG leads. By combining Physics-Informed Neural Networks and Generative Adversarial Networks (GAN), the authors propose a GAN model with PDE loss. On the Georgia 12-Lead ECG Challenge dataset, the authors evaluate the model's performance using specificity as a metric and believe they have surpassed the state-of-the-art.

**Strengths:**

1. Applying PINN onto ECG generation is a great approach. Using traditional ECG modeling as regularization can effectively control the generation of the model, which makes sense.

2. This work achieved state-of-the-art results on specificity.

3. This paper is very readable, clearly written, and easy to follow.

**Weaknesses:**

1. Model validation is a significant issue:

i) The dataset the authors used consists of 10,344 12-lead ECG recordings obtained from 7,871 patients. This means that a patient can contribute to more than one case. The authors do not seem to clarify whether a patient appears more than once in both the training and testing sets during data partitioning.

ii) There is a wealth of open-source ECG data available; it is unclear why the authors only used one dataset.

iii) The generated results have numerous validation metrics, not just specificity, such as 1-NNC and rFID. Additionally, shouldn't some visualization results be presented and analyzed?

iv) Many key state-of-the-art methods are not compared, such as ME-GAN `[1]` and DiffuSETS `[2]`, and there is also no discussion of them.

2. Regarding the innovation: Combining the physical model of ECG with GAN is not a particularly novel idea, as there has been considerable exploration in this area, even without claiming it with the concept of PINN. I did not find any distinct technical or application innovations in the paper.

Ref:

`[1]` ME-GAN: Learning Panoptic Electrocardio Representations for Multi-view ECG Synthesis Conditioned on Heart Diseases, ICML

`[2]` DiffuSETS: 12-lead ECG Generation Conditioned on Clinical Text Reports and Patient-Specific Information, KDD

**Questions:**

See Weakness.

---

### Official Review · Reviewer_7kxi · 2024-11-01

**Soundness:** 3
**Presentation:** 3
**Contribution:** 3
**Rating:** 6
**Confidence:** 4

**Summary:**

The paper introduces PDECGAN, an innovative generative framework that combines partial differential equations (PDEs) with a generative adversarial network (GAN) to synthesize realistic 12-lead ECG signals. By embedding PDE-based constraints into the GAN architecture, PDECGAN captures both the temporal evolution and spatial relationships inherent to cardiac electrophysiology. Experiments demonstrate that data augmentation with synthetic data generated by PDECGAN effectively improves model classification performance, surpassing advanced comparison methods.

**Strengths:**

1. The incorporation of partial differential equations (PDEs) into the generative framework is highly novel, enabling the model to synthesize ECGs with realistic physiological significance.
2. The experiments provide comprehensive comparisons with state-of-the-art generative methods, demonstrating the superior performance of the proposed approach.
3. By adjusting the parameter λPDE, the study effectively validates the performance balance between temporal and spatial components, showcasing the model’s flexibility in capturing spatiotemporal dynamics.

**Weaknesses:**

1. The paper lacks figures, such as diagrams illustrating the model architecture and examples showcasing the quality of synthesized ECG signals.
2.In the experimental section, only specificity changes are tested while keeping sensitivity fixed, thus only evaluating the model's ability to recognize negative classes. Including additional metrics, such as AUROC (used in SSSD-ECG), could better reflect the overall classification performance.
3. Beyond classification performance, it would be valuable to include additional experiments or metrics that assess the model's ability to "accurately model the spatiotemporal dynamics of 12-lead ECG signals" and "captures the complex relationships between leads." For instance, analyzing morphological details of the synthesized signals or having medical experts evaluate the realism of the data would provide further insights into the model's effectiveness.

**Questions:**

1. When splitting the dataset, "dividing the dataset into training and validation subsets … 20% of the dataset was set aside as an independent test set," does this ensure that samples from the same recording are not placed in both the training and testing sets?
2. For segmenting signals by heartbeat cycles, do different signals have different “L” values? Additionally, could the authors provide more information on the training procedure, such as batch size, number of training epochs?
3. Does the model directly synthesize 12-lead signals, and if so, do the synthesized signals comply with the inter-lead relationships described in Appendix A?

---

### Official Review · Reviewer_awJ1 · 2024-11-04

**Soundness:** 3
**Presentation:** 3
**Contribution:** 2
**Rating:** 3
**Confidence:** 4

**Summary:**

This paper introduces PDECGAN, a novel approach to generate high-quality synthetic electrocardiogram (ECG) data to support machine learning model development for heart disease diagnosis. ECG data inherently involves both temporal and spatial variability, making the realistic generation of 12-lead ECGs challenging for traditional generative models. To address these challenges, the authors propose integrating partial differential equations (PDEs) into a GAN framework, termed PDECGAN. This model mathematically represents the heart’s electrical activity, thereby adhering to physiological constraints and capturing the interdependencies between the 12 leads to produce realistic and reliable synthetic ECG signals.

**Strengths:**

PDECGAN effectively models physiological constraints and accurately reflects the interactions between the 12 leads, resulting in synthetic ECG data that is more accurate and reliable than that produced by previous models. This fidelity not only enhances the performance of diagnostic models but also improves compatibility with various neural network architectures, while naturally capturing temporal variations. Moreover, this approach addresses ethical concerns and data scarcity issues associated with real patient data, a significant advantage in synthetic medical data generation. Experimental results also demonstrate that classifiers trained on PDECGAN-generated data show improved detection of specific heart abnormalities, thereby contributing to model robustness.

**Weaknesses:**

The rationale for using PDEs to enforce physiological constraints is somewhat underexplored. Existing models like the ODE-based ECG ODE-GAN and VCG-utilizing 3KG model have successfully recreated ECG data by capturing physiological principles. A more thorough explanation of how PDECGAN differentiates itself from these methods, as well as the intuitive and practical advantages of using PDEs, would be beneficial. If PDEs indeed provide superior advantages over previous methods, these should be clearly articulated with supporting evidence. Additionally, experimental validation comparing the performance of PDECGAN with existing models would be helpful in demonstrating these differences concretely.

**Questions:**

The paper claims that training with synthetic data generated by PDECGAN improves model performance, but experimental results indicate that as the amount of data (N) increases, performance does not rise linearly. This suggests that adding synthetic data may lead to overfitting on certain patterns rather than capturing the full variability of the data. Additionally, due to the inherent randomness of generated data, increasing data volume may lead to distributional discrepancies from real data beyond a certain threshold. Further analysis or experiments could clarify the non-linear relationship between data volume and performance.

Furthermore, since chest leads can reportedly be generated through vector calculations, it would be interesting to see how PDECGAN’s performance compares against this approach. Adding experimental results or discussion on this comparison would add valuable insights into the model’s relative performance.

---

### Note · Authors · 2024-11-25

**Comment:**

We sincerely thank the reviewers for their time and thoughtful feedback on our paper. We appreciate the constructive insights provided. Given the limited time for the rebuttal process, we have decided to withdraw our submission to focus on implementing these suggestions and preparing a more robust version of the paper for future submission. We are grateful for the reviewers’ efforts and look forward to presenting an improved version of our work in the near future.

**Withdrawal Confirmation:**

I have read and agree with the venue's withdrawal policy on behalf of myself and my co-authors.